# Efficient Light Coupling and Purcell Effect Enhancement for Interlayer Exciton Emitters in 2D Heterostructures Combined with SiN Nanoparticles

**DOI:** 10.3390/nano13121821

**Published:** 2023-06-08

**Authors:** Alexandra D. Gartman, Alexander S. Shorokhov, Andrey A. Fedyanin

**Affiliations:** Faculty of Physics, Lomonosov Moscow State University, Moscow 119991, Russia; gartman@nanolab.phys.msu.ru (A.D.G.); shorokhov@nanolab.phys.msu.ru (A.S.S.)

**Keywords:** integrated photonics, Mie resonances, 2D heterostructures, interlayer excitons

## Abstract

Optimal design of a silicon nitride waveguide structure composed of resonant nanoantennas for efficient light coupling with interlayer exciton emitters in a MoSe${}_{2}$–WSe${}_{2}$ heterostructure is proposed. Numerical simulations demonstrate up to eight times coupling efficiency improvement and twelve times Purcell effect enhancement in comparison with a conventional strip waveguide. Achieved results can be beneficial for development of on-chip non-classical light sources.

## 1. Introduction

Two-dimensional (2D) materials, ranging from semimetallic graphene to insulating hexagonal boron nitride, have attracted significant interest from the scientific community recently [1,2,3]. In comparison with conventional bulk materials, these atomically thin layers exhibit exceptional optical properties such as enhanced nonlinear optical response and ultrafast photo-induced relaxation [4,5,6]. Transition metal dichalcogenide (TMD) materials, which can be represented as MX${}_{2}$ (M = Mo, W; X = S, Se), are well-renowned in this research field for their electronic and optical properties [7,8,9,10,11,12]. Contrary to gapless graphene, these materials undergo a change from indirect to direct bandgap semiconductors being thinned down to a monolayer level, which corresponds to the quantum confinement effect. They also possess large exciton binding energies defining their remarkable optical properties. The bandgap energy in TMD thin films is spanned across the visible and the near-infrared spectral ranges making them attractive for variety of photonic and optoelectronic devices [13].

Owing to their fascinating properties, 2D TMD materials can be used for many practical applications such as optical modulators, classical and non-classical light sources, light-harvesting devices, photodetectors [14,15,16,17,18,19,20]. Due to their intrinsic thickness light extraction efficiency from TMD films can significantly outperform other solid-state emitters [21,22]. It makes them especially interesting in terms of integration with different photonic structures and resonant cavities [23]. Moreover, TMD materials can provide a versatile platform for spin–photon interfaces in the field of quantum networking [24].

Monolayers of different TMD materials also can be vertically stacked to form heterostructures with new peculiar properties, which are not observed in single layers of constituent materials [25]. One of the emerging effects is the formation of interlayer excitons corresponding to spatially separated electron–hole pairs [26]. They are known for their long lifetimes [27] and associated dipole moment orientation in the out-of-plane direction, which both can be beneficial for the creation of deterministic quantum emitters [28]. Together with the intrinsic long lifetime, valley degree of freedom and tunability make interlayer excitons a promising candidate for a broad range of quantum photonic applications [29].

In order to fully utilize their functionality for existing devices, TMD layers can be optically coupled with photonic structures on a chip [30]. It opens a way for different practical devices such as sensors, radiofrequency systems and optoelectronic circuits [31]. Another important application for hybrid integrated photonic structures can be found in the field of quantum optics [32]. Integration of single photon emitters in a TMD monolayer with the guided mode of a silicon nitride waveguide has been also demonstrated recently [33].One of the main challenges, which significantly decreases light coupling efficiency in such systems, is the unpredictable orientation of the dipole polarization corresponding to the localized exciton emitter in the TMD monolayer. Moreover, the evanescent field of the waveguide mode has a limited ability to channel radiated power even for the fixed dipole orientation estimated below 10% for the presented case [33]. Employment of resonant nanoantennas for emitters in 2D layered materials can bring about both improved optical coupling and Purcell effect enhancement [34,35,36].

Resonant integrated photonics based on Mie-type nanoparticles has recently been introduced as a versatile tool to control light on a chip [37,38]. It can provide compactness and new functionalities not shown previously in conventional components. By tailoring the resonant response of nanoparticles, it is possible not only to guide radiation through them [39,40], but also to improve light coupling from the external source to a chip [41]. Resonant nanophotonic cavities for lasing and all-optical modulation also have been demonstrated previously [42,43]. However, the major part of these works deals with electric-type modes, which cannot be efficiently coupled with out-of-plane dipole sources on a chip. Through special design of the nanoparticles, one can achieve both high density of optical states in the cavity and radiation channeling via strong near-field connection.

In this paper, we propose a concept of a resonant waveguide system (RWS) based on a magnetic type high-Q photonic nanocavity for effective optical coupling with localized interlayer exciton emitters in vertically stacked monolayer MoSe${}_{2}$–WSe${}_{2}$ heterostructures, as conceptually illustrated in Figure 1a. The RWS is composed of a silicon nitride (SiN) strip waveguide, the central part of which is replaced by rectangular nanoparticles forming specially designed subwavelength Bragg grating. Silicon nitride is chosen as it is a typical material for integrated silicon photonics and available for large-scale fabrication through the world-known factories [44]. It has the transparency region in visible and near-infrared (NIR) spectral ranges, making it suitable for many applications including quantum optics [45]. It is also a good match for the present study contrary to silicon used in the relative previous work [41] due to the typical emission spectra of TMD films. The proposed RWS can significantly increase both the spontaneous emission rate (Purcell effect) and optical coupling of the interlayer exciton emitters in TMD heterostructures with the integrated waveguide due to the effective mode matching under Mie-type magnetic dipole excitation in the constituent nanoparticles.

## 2. Materials and Methods

The RWS under study is composed of the nanoparticle chain placed between two strip silicon nitride waveguides with the height of *h* = 400 nm and the width of *w* = 850 nm on a silica substrate (Figure 1a). The whole chain consists of three sections: left and right subwavelength Bragg-gratings (SBG) formed by 10 equal identical nanoparticles with the same height and width as the strip waveguide and the length of *l* = 243 nm and the central nanocavity formed by 20 nanoparticles with the varying length (Figure 1c). The period of nanoparticles in the chain equals *p* = 297 nm and is optimized to obtain a photonic stop band in the spectral region of the interlayer exciton emission in monolayer MoSe${}_{2}$–WSe${}_{2}$ heterostructures [46]. The total length of the SBG section is $L_{bg}$ = 2.97 μm.

The central nanocavity section includes twenty particles of varying sizes and has the total length of $L_{c}$ = 5.94 μm. The length of nanoparticles in this part follows a parabolic profile to keep the high-Q factor of the resonator [47]. It changes from 243 nm on the edges to 205 nm in the center of the cavity. As a result, an optimal Gaussian-shaped field attenuation profile is achieved and the nanocavity resonance peak coincides with the emission line of the interlayer exciton emission in monolayer MoSe${}_{2}$–WSe${}_{2}$ heterostructures [46]. The above-listed parameters are obtained through the following procedure. At the first step, we optimize the strip waveguide size for the first TM-mode propagation in the desired spectral range and then calculate its effective refractive index for the target wavelength corresponding to the interlayer exciton transition line. At the second step, we use this waveguide effective refractive index to find a subwavelength Bragg-grating period suitable to support photonic stop-band in the desired spectral range. Then, for geometrical parameters of nanoparticles in the central part of the cavity, we employ a parabolic modulation profile to achieve a defect state with a high quality factor in the photonic stop-band as it is highlighted above (see [48]). Finally, the RWS parameters are tuned to match the interlayer exciton transition spectral position for the chosen TMD heterostructure.

The aspect ratio (height/width) of the nanoparticles is chosen in analogy with Ref. [40] to support magnetic Mie-type dipole resonance excitation in the spectral vicinity of the interlayer exciton emission in monolayer MoSe${}_{2}$–WSe${}_{2}$ heterostructures [46]. The finite difference time domain (FDTD) method realized in the commercial software Ansys Lumerical FDTD is used to calculate a transmission spectrum of the RWS (Figure 1b). For this purpose, we use a fundamental TM mode source placed in the left strip waveguide section and a power monitor located in the right waveguide section after the nanoparticles chain. SiN refractive index dispersion for simulations is obtained by the ellipsometry measurements of the thin films (with the height of 350 nm) grown by the PECVD method with the consequent annealing. The input TM mode profile in the strip waveguide cross-section is shown in the inset of Figure 1b. The transmission spectrum reveals the cavity resonance peak at the $\lambda_{res}=918\mathrm{nm}$ with the quality factor $Q∼10^{3}$ similar to previous works [39].

Figure 1c shows electrical near-field distribution in the middle cross-section of the RWS at the resonant wavelength. The field is strongly confined in the center of the cavity and provides a higher density of states at a fixed resonant wavelength corresponding to inter-layer exciton transition in the TMD heterostructure. It is worth mentioning that, contrary to previous works ([39]), we deliberately optimize our structure for TM mode configuration with a dominant Ey electric field component corresponding to the out-of-plane orientation of interlayer excitons in the TMD heterostructure. This provides better optical coupling between associated with it emitters and the RWS. It is worth mentioning that the designed structure can be fabricated by standard techniques such as electron-beam lithography and reactive ion etching. The TMD heterostructure can be further transferred to the fabricated structure using the mechanical exfoliation method.

## 3. Results

In order to analyze the emission enhancement and radiation coupling efficiency associated with interlayer localized excitons in stacked MoSe${}_{2}$–WSe${}_{2}$ monolayers combined with designed RWS, we provide a series of numerical simulations with varying positions of the dipole emitter in the heterostructure.

### 3.1. Analyzing Value of the Parcel Effect Depending on the Position of the Dipole Emitter in the TMD Film

First, we trace the dependence of the Purcell effect and optical coupling efficiency on the position of the dipole emitter along the RWS. For Purcell effect enhancement definition, we introduce a factor $A_{eff}=P/P_{0}$, where *P* is radiated dipole power in the presence of the RWS and $P_{0}$ is the power that the same dipole would radiate in vacuum since the emission rate is proportional to the local density of optical states (LDOS), and the LDOS is proportional to the power emitted by the source. Figure 2a shows the dependence of enhancement factor $A_{eff}$ on the emission wavelength for various dipole locations within the structure depicted by different colors in the inset (single step equals to 1.8 μm). Maximum value of $A_{eff}=16$ corresponds to the case when the out-of-plane electric dipole is localized above the central part of the RWS (central part of the nanocavity). It coincides with the electric field hotspot providing the maximum density of optical states. The inset in Figure 2a demonstrates dipole power transmission in one particular direction of the RWS. One can note that almost 35% of the total emitted radiation is transmitted through the RWS into the strip SiN waveguide section. It is eight times larger in comparison with the conventional SiN strip waveguide with the same parameters (width and height), which emphasizes the superior optical coupling efficiency of the designed RWS. One can also compare these results with similar examples in the literature. For different material platforms (hBN and monolayer TMD films on the integrated photonic chips) demonstrated optical coupling efficiency and Purcell effect enhancement are inferior compared to the designed RWS [33,49,50,51].

The Purcell factor as well as optical coupling efficiency decrease with the dipole shifting along the nanoparticle chain. $A_{eff}$ drops four times while transmission drops by around 10% for dipole movement from the center of the RWS to its edge. At the same time, both effects still prevail over the conventional strip waveguide case by at least four times.

Figure 2b shows normalized electric field y-component (top) and magnetic field amplitude (bottom) distributions for xz and zy middle cross-sections through one of the central RWS nanoparticles when the dipole source is located right above it. This out-of-plane dipole associated with the localized interlayer exciton in the MoSe${}_{2}$–WSe${}_{2}$ heterostructure initiates magnetic Mie-type resonance in the central SiN nanoparticle, as can be seen in the right part of Figure 2b. It supports a strong optical connection between the emitter and the RWS, providing both enhanced $A_{eff}$ and radiation transmission through the waveguide chain.

### 3.2. Transverse Shift of the Emitter in the Subwavelength Waveguide Chain

We also examine the dependence of the Purcell effect and the optical coupling efficiency on the dipole emitter transverse shift relative to the RWS. Two cases are considered: the dipole is in the center and on the edge of the RWS (see Figure 3). These are the most possible positions of localized excitons in transferred 2D films due to the mechanical strain [33,52].

The spectra of the radiation power transmitted along the RWS and the spectra of the A${}_{eff}$ coefficient are shown by solid curves in Figure 3 for the case of dipole emitter located in the center (Figure 3a) and on the edge (Figure 3b) of the RWS. Similar results for the conventional strip waveguide are shown by dashed curves. In the vicinity of the RWS resonant wavelength $\lambda_{res}$, the optical coupling efficiency for both transverse dipole positions is more than 7 times larger for the RWS relative to the conventional strip waveguide. Results for the A${}_{eff}$ coefficient demonstrate up to 12 times the Purcell effect enhancement in the center of the RWS compared to the conventional strip waveguide, while for the edge position it drops almost three times due to the local electric field decrease. Still, it is four times larger for the RWS relative to the conventional strip waveguide, making it a superior choice for effective 2D emitter integration.

We also perform simulations to estimate the optical coupling efficiency between intralayer exciton emitters (with in-plane orientation) and the conventional strip waveguide (configuration similar to the work [33]). Both longitudinal and transverse orientations are considered. For the first case, the transmission reaches 9.3%, while for the second case it equals to 8.9%. These results are up to four times smaller in comparison with the RWS presented above combined with interlayer exciton emitters in the TMD heterosctructure.

It is worth mentioning that the Purcell effect increases the spontaneous emission rate of excitonic emitters due to a much higher photon density of states in the nanocavity. Typical lifetimes for interlayer excitons in TMD heterostructures are in the order of a few tens of ns [53]. Taking into consideration the Purcell factor of 16 (obtained for the best configuration in the designed RWS), one can expect the interlayer exciton lifetime of a few ns, which is larger compared to intralayer excitons in single TMD monolayers [54]. Considering this effect in the perspective of single-photon emitters, localized interlayer excitons can provide a much narrower spectral linewidth, which is beneficial in terms of photon indistinguishability.

Finally, for the experimental validation of the proposed system, one needs to take into consideration the variation of the RWS geometrical parameters during the sample fabrication. To this end, we have estimated the possible resonance shift and broadening for random geometrical parameter variations typical for electron-beam lithography and reactive ion etching processes. The results demonstrate that the resonance quality factor is not significantly affected, while the spectral shift in real devices can be compensated, for example, by electrical tuning of the TMD heterostructure [53].

## 4. Conclusions

We propose the optimized resonant waveguide system composed of SiN nanoparticles forming magnetic type high-Q photonic nanocavity for efficient integration with interlayer exciton emitters in vertically stacked monolayer MoSe${}_{2}$–WSe${}_{2}$ heterostructures. Obtained numerical results demonstrate up to eight times optical coupling efficiency improvement and up to twelve times Purcell effect enhancement in comparison with the conventional strip waveguide. Our findings open prospects for a variety of applications such as integrated single-photon emitters based on long-living excitonic states in TMD heterostructures.

## Figures and Tables

**Figure 1 nanomaterials-13-01821-f001:**
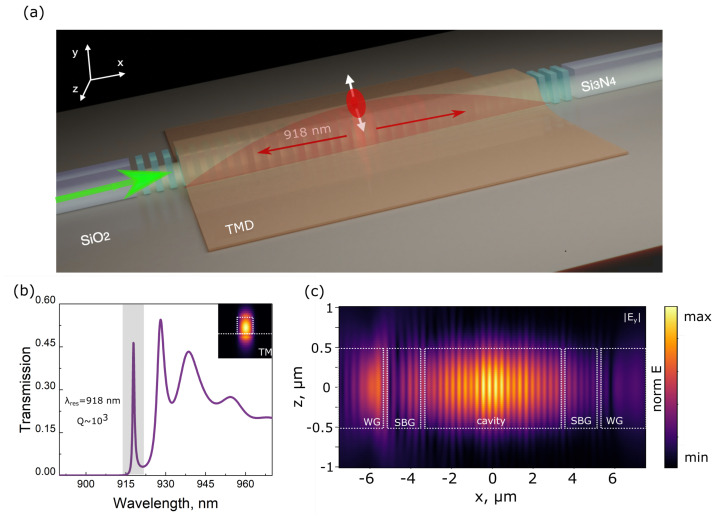
(**a**) Schematic illustration of the optical coupling between the resonant waveguide structure and the localized interlayer exciton source in vertically stacked monolayer TMD films. (**b**) RWS transmission spectrum obtained for the fundamental TM mode source placed in the strip waveguide section. The inset demonstrates near-field distribution of the TM mode in the waveguide cross-section at resonant wavelength $\lambda_{res}=918\;\mathrm{nm}$. (**c**) Top-view of the electrical near-field distribution in the middle cross-section of the RWS at the resonant wavelength of $\lambda_{res}=918\;\mathrm{nm}$.

**Figure 2 nanomaterials-13-01821-f002:**
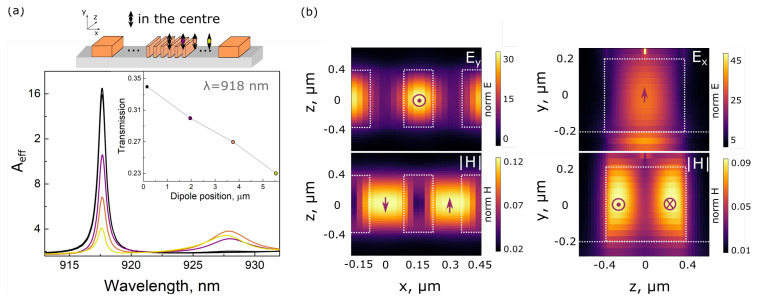
(**a**) Spectra of the A${}_{eff}$ coefficient for the various dipole emitter location along the RWS as illustrated by the top scheme. The inset shows transmission results for the same dipole positions. (**b**) Normalized electric field y-component (top) and magnetic field amplitude (bottom) distributions for xz and zy middle cross-sections through the central RWS nanoparticles.

**Figure 3 nanomaterials-13-01821-f003:**
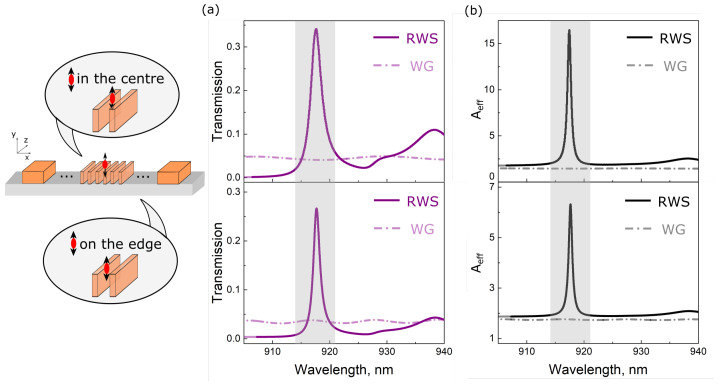
Transmission and A${}_{eff}$ coefficient spectra for the dipole emitter placed in the center (**a**) and on the edge (**b**) of one of the central RWS nanoparticles. Results for the conventional strip waveguide are depicted by dashed curves.

## Data Availability

Not applicable.

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
