# Peer review of "Efficient Light Coupling and Purcell Effect Enhancement for Interlayer Exciton Emitters in 2D Heterostructures Combined with SiN Nanoparticles"

_nanomaterials, 2023, doi:10.3390/nano13121821_

Round 1

Reviewer 1 Report

This paper presents an optimized heterostructures that integrates two-dimensional materials, demonstrating its potential in enhancing light coupling efficiency and Purcell effect in typical photonic devices. While the research holds promise and the topic is of interest, I recommend revisions to strengthen its clarity and impact.

 Suggested revisions:

 1.       In the introduction, please provide more detailed information about the significance of 2D materials and their integration with typical photonic devices to help readers better understand their importance in practical applications.

 2.       When describing the design of the heterostructures, please consider providing more details about the optimization process and parameter selection to help readers better understand the design principles and advantages.

3    3. When comparing the proposed structure to the conventional structure, please add appropriate references to support the claims made and to highlight the novelty and advantages of the new approach.

4   4.  In the conclusion, please summarize the challenges and limitations that the proposed structure may face in practical applications and suggest potential solutions or improvements.

 5. The paper's conclusion mentions the integration of single-photon emitters based on long-lived excitonic states. It would be helpful if the authors could briefly discuss the potential influence of the Purcell effect on the excitonic lifetime, as the effect could lead to a significant reduction in lifetime due to the enhanced radiative decay rate.

Author Response

Reply to Reviewers’ comments

and a summary of the changes made in the revised manuscript

Reviewer 1:

“In the introduction, please provide more detailed information about the significance of 2D materials and their integration with typical photonic devices to help readers better understand their importance in practical applications.”

Our response:

We thank the reviewer for his/her comment. We’ve added in the introduction section more details about 2D materials integration with photonic devices and emphasized its importance for practical applications.

Text added: (p.1, line 26) Moreover, TMD materials can provide a versatile platform for spin–photon interfaces in the field of quantum networking [10.1038/s42254-021-00408-0].

Text added: (p.1, line 33) Together with the intrinsic long lifetime, valley degree of freedom and tunability make interlayer excitons a promising candidate for a broad range of quantum photonic applications [10.1038/s41467-021-23732-6].

Text added: (p.1, line 37) It opens a way for different practical devices such as sensors, radiofrequency systems and optoelectronic circuits [10.1038/s41928-021-00672-z]. Another important application for hybrid integrated photonic structures can be found in the field of quantum optics [10.1002/adom.201901132].

Reviewer 1:

“When describing the design of the heterostructures, please consider providing more details about the optimization process and parameter selection to help readers better understand the design principles and advantages.”

Our response:

We thank the Reviewer for this valuable comment. To clarify our optimization process and parameter selection we’ve added more details about the RWS numerical modeling to the manuscript.

  • At the first step we optimize strip waveguide parameters to find the aspect ratio (height/width) suitable for the first TM-mode propagation in the desired spectral range and then calculate its effective refractive index for the target wavelength corresponding to the interlayer exciton transition line.
  • At the second step we use this waveguide effective refractive index to find a subwavelength Bragg-grating period suitable to support photonic stop-band in the desired spectral range.
  • At the next step, for geometrical parameters of nanoparticles in the central part of the cavity we employ a parabolic modulation profile to achieve a defect state with a high quality factor in the photonic stop-band as it’s shown in the work 1364/OE.18.015859 (W_x(i)=W_x0(1+(i/m)2).
  • Finally, the RWS parameters are tuned to match interlayer exciton transition spectral position for the chosen TMD heterostructure (stack of MoSe_2–WSe_2 monolayers).

In addition, to provide better coupling efficiency between the emitter and the RWS we optimize SiN nanoparticles to direct their scattering profile maxima into the waveguide section (see Fig. R1). It allows for the effective propagation of radiation energy through the RWS and further into the photonic chip.

Fig. R1 Scattering cross-section results for two cases corresponding to the dipole emitter located above the central nanoantenna (a) and the edge nanoantenna (b) in the RWS.

Text added: (p.2 , line 89) Listed above parameters are obtained through the following procedure. At the first step we optimize the strip waveguide size for the first TM-mode propagation in the desired spectral range and then calculate its effective refractive index for the target wavelength corresponding to the interlayer exciton transition line. At the second step we use this waveguide effective refractive index to find a subwavelength Bragg-grating period suitable to support photonic stop-band in the desired spectral range. Then for geometrical parameters of nanoparticles in the central part of the cavity we employ a parabolic modulation profile to achieve a defect state with a high quality factor in the photonic stop-band as it is highlighted above (see 10.1364/OE.18.015859). Finally, the RWS parameters are tuned to match interlayer exciton transition spectral position for the chosen TMD heterostructure.

Reviewer 1:

“When comparing the proposed structure to the conventional structure, please add appropriate references to support the claims made and to highlight the novelty and advantages of the new approach.”

Our response:

We thank the reviewer for this comment. We’ve added more references about coupling of localized 2D emitters with integrated photonic chips (supplementing the reference 10.1038/s41467-019-12421-0 already highlighted in the manuscript):

1) (10.1364/OME.9.000441) This work reports coupling of emitters in a monolayer TMD film with a lithium niobate photonic chip. The coupling efficiency is well below values obtained in our work.

2) (10.1021/acs.nanolett.2c03151) In this paper authors integrate emitters in 2D films (hBN and TMD) with SiN microresonators. While they demonstrate similar values of the coupling efficiency (considering 2 waveguide directions in our system), the Purcell factor is an order of magnitude smaller compared to the RWS designed in our work.

3) (10.1021/acsnano.0c04397) In this paper authors integrate emitters in 2D hBN films with SiN waveguides. Achieved coupling efficiency a few times smaller compared to the RWS designed in our work.

Text added: (p. 4, line 144) One can also compare these results with existing in the literature similar examples. For different material platforms (hBN and monolayer TMD films on the integrated photonic chips) demonstrated optical coupling efficiency and Purcell effect enhancement are inferior compared to the designed RWS [10.1038/s41467-019-12421-0, 10.1364/OME.9.000441, 10.1021/acs.nanolett.2c03151, 10.1021/acsnano.0c04397].

Reviewer 1:

“In the conclusion, please summarize the challenges and limitations that the proposed structure may face in practical applications and suggest potential solutions or improvements.”

Our response:

We thank the Reviewer for this valuable suggestion. We’ve added information about possible experimental issues and different ways to address them. Let us summarize it below.

1) First, as it is highlighted in the manuscript, in the experiment we can't predict the exact position of localized excitons in the structure. From literature it is known that there are two typical positions of localized excitons in 2D films transferred on to the integrated waveguide due to the mechanical strain – in the center and on the edge of the waveguide. We consider both cases in our modeling and demonstrate up to 4 times improvement for the designed RWS even in the worst possible case compared to the conventional strip waveguide. However, special strain engineering prior to the 2D film transfer can help to gain maximum efficiency (see for example 10.1126/sciadv.1701696).

2) The second thing we need to take into consideration is the variation of the RWS geometrical parameters during the sample fabrication. It can lead to the resonance spectral position shift and its quality factor decrease. For experimental validation we can mitigate this problem by the electron beam dose variation during the lithography as well as through parameter variation in the experimental sample layout (see Fig. R2 for example). For practical devices one can also consider electrical tuning of the TMD heterostructure to match the interlayer exciton line with the RWS resonance position (see for example 10.1126/sciadv.1701696).

Fig. R2 The experimental layout of  (a) one set of RWS structures with variation of the width and the number of nanoantennas in the structure central part and (b) single RWS with optimized by numerical simulations parameters.

It is worth mentioning that we’ve also estimated the possible resonance shift and broadening for random geometrical parameters variations (typical for electron-beam lithography and reactive ion etching processes). Results are shown in Fig. R3. We can see that the quality factor is not significantly affected, while the spectral shift can be compensated by the methods discussed above.

Fig. R3 Calculated transmission spectra for the ‘ideal’ RWS and for the case of random geometrical parameters variations.

Text added: (p. 5, line 183) Finally, for the experimental validation of the proposed system one need to take into consideration the variation of the RWS geometrical parameters during the sample fabrication. To this end we’ve estimated the possible resonance shift and broadening for random geometrical parameters variations typical for electron-beam lithography and reactive ion etching processes. Results demonstrate that the resonance quality factor is not significantly affected, while the spectral shift in real devices can be compensated, for example, by electrical tuning of the TMD heterostructure [10.1126/sciadv.1701696].

Reviewer 1:

“The paper's conclusion mentions the integration of single-photon emitters based on long-lived excitonic states. It would be helpful if the authors could briefly discuss the potential influence of the Purcell effect on the excitonic lifetime, as the effect could lead to a significant reduction in lifetime due to the enhanced radiative decay rate”

Our response:

We are grateful for this question. Indeed, we agree that the Purcell effect influences the excitonic lifetime. Particularly, it increases the spontaneous emission rate of excitonic emitters due to a much higher photon density of states in the nanocavity. The Purcell factor can be expressed as Fp = (3/4π2)(Q/V)(λ0/n)3 and the total lifetime is determined by the harmonic mean of the non-radiative (t_nr) and radiative (t_rad) lifetime:  1/t=1/(t_nr)+1/(t_rad). Typical lifetime for interlayer excitons in TMD heterostructures are in the order of a few tens of ns (see for example 10.1126/sciadv.1701696). Even taking into consideration the Purcell factor of 16 (obtained for the best configuration in our system) we still have the lifetime of a few ns, which is much larger compared to intralayer excitons (see for example 10.1103/PhysRevB.96.155423). Considering this effect in the perspective of single-photon emitters, localized interlayer excitons will provide narrower spectral linewidth, which is beneficial in terms of photon indistinguishability.

Text added: (p. 5, line 191) It is worth mentioning that the Purcell effect increases the spontaneous emission rate of excitonic emitters due to a much higher photon density of states in the nanocavity. Typical lifetime for interlayer excitons in TMD heterostructures are in the order of a few tens of ns [10.1126/sciadv.1701696]. Taking into consideration the Purcell factor of 16 (obtained for the best configuration in the designed RWS) one can expect the interlayer exciton lifetime of a few ns, which is larger compared to intralayer excitons in single TMD monolayers [10.1103/PhysRevB.96.155423]. Considering this effect in the perspective of single-photon emitters, localized interlayer excitons can provide much narrower spectral linewidth, which is beneficial in terms of photon indistinguishability.

Reviewer 2 Report

The authors present a numerical simulation for the light coupling and Purcell enhancement in the near-infrared. This kind of approach has been extensively reported in literature and there are not relevant novelties in the current manuscript. In addition to this, the manuscript present several issues.

Firstly, it is quite disapoiting when the reader expect for an heterostructure coupled to a semiconductor waveguide while the authors provide only and approach operating at the same wavelength. Secondly, the authors do not discuss about the posibilities for fabricating their proposal.  And secondly, the lack of an experimental realization makes this work useless for the community. As a result I should recomment to reject this manuscript.

I believe that the experimental realization of the approach could be of interest for nanomaterials readers. In its current state I would reduce the claim for of the 2D heterostructure and try with a journal of optics or engeneering.

Hoping to be helpful.

Kind regards

Author Response

Reviewer 2:

The authors present a numerical simulation for the light coupling and Purcell enhancement in the near-infrared. This kind of approach has been extensively reported in literature and there are not relevant novelties in the current manuscript.

Our response:

We thank the Reviewer for his/her comment. Indeed, there are plenty of works dealing with integrated photonic cavities and dipole emitters in different systems ranging from nanodimonds to 2D materials. However, the main problem for all the studied to date examples is the orientation misalignment between the dipole emitter polarization and the cavity optical mode configuration. It is thoroughly investigated, for example, in the work 10.1038/s41467-019-12421-0, where authors state that due to the random orientation of the in-plane exciton emitters in WSe2 their radiation coupling efficiency is significantly decreased. At the same time, interlayer excitons in 2D heterostructures are well known for their distinct out-of-plane orientation, but have never been proposed before for solving this problem (to the best of our knowledge). Combination of such emitters with specifically designed nanophotonic cavities can provide significant improvement of the radiation on-chip coupling. However, previously investigated photonic nanocavities typically have in-plane electric field configuration and cannot be efficiently coupled with such emitters. In this work for the first time we design a magnetic type nanophotonic cavity based on SiN nanoparticles to provide not only high-Q resonance for the Purcell effect enhancement but also for highly effective radiation coupling to the photonic chip by thoroughly shaping SiN nanoparticles scattering diagram. To the best of our knowledge, there aren't any works considering the combination of all these factors. As a result, our designed structure has up to 8 times optical coupling efficiency improvement and up to 12 times Purcell effect enhancement in comparison with the conventional systems.

Reviewer 2:

Firstly, it is quite disapoiting when the reader expect for an heterostructure coupled to a semiconductor waveguide while the authors provide only and approach operating at the same wavelength.

Our response:

We thank the Reviewer for his/her comment. Unfortunately, we can’t grasp it fully, but we’ll try to address it following our understanding. If the reviewer means the wavelength limitation, then certainly our RWS can be easily scaled for other wavelengths in the NIR spectral region. Changing the ratio between RWS height and width we can tune the spectral position of the resonant mode. At the same time in our work we focus on specific luminescence data for the TMD heterostructure and optimize response of the RWS in accord with it. Moreover, for experimental structures with fixed parameters spectral position of the TMD excitonic emission lines can be tuned by electrical biasing or strain engineering as was shown in the previous works (see for example 10.1126/sciadv.1701696).

Reviewer 2:

Secondly, the authors do not discuss about the posibilities for fabricating their proposal.  And secondly, the lack of an experimental realization makes this work useless for the community.

Our response:

We thank the reviewer for pointing out this issue. It’s worth mentioning that we have already started the fabrication process for experimental validation. Moreover, our modeling is based on real ellipsometry data measured on thin SiN films, which we plan to use for experimental sample fabrication. The sample layout for resistive mask creation in accord with the obtained by modeling geometrical parameters has been also created (see Fig. R2 above). We plan to use electron-beam lithography followed by reactive ion etching to create SiN nanophotonic structures. After that the mechanical exfoliation method will be applied to transfer TMD monolayers onto the structure. We’ve already worked out this procedure for the similar structures. Following the reviewer's comment we’ve supplemented our manuscript with additional information about possible fabrication workflow. However, this whole fabrication process takes time and cannot be finished in the near future. We believe that experimental validation itself deserves an independent publication, while presented in the current manuscript numerical study can be also interesting for a broad community of researchers (please check our discussion about novelty above).

Text added: (p. 4, line 110) It is worth mentioning that the designed structure can be fabricated by standard techniques such as electron-beam lithography and reactive ion etching. The TMD heterostructure can be further transferred to the fabricated structure using the mechanical exfoliation method.

Reviewer 3 Report

The paper is relatively well written and seems technically sound. My main concern is the lack of technical detail, namely, the way the enhancement factor A_eff was computed (Fig. 2a). In particular, what was the procedure for calculating the radiated power? It seems that the authors simply used the local electric field norm squared. I guess it can be justified but some discussion would be helpful. 

Some moderate copy editing is required.

Author Response

Reviewer 3:

“ My main concern is the lack of technical detail, namely, the way the enhancement factor A_eff was computed (Fig. 2a). In particular, what was the procedure for calculating the radiated power? It seems that the authors simply used the local electric field norm squared. I guess it can be justified but some discussion would be helpful. ”

Our response:

We thank the reviewer for his/her positive feedback on our work. Following his question we’ve supplemented our manuscript with additional information about how the A_eff was computed. First of all, we want to emphasize that it’s not simply the local electric field norm squared as it was guessed by the reviewer. The Purcell factor is the emission rate enhancement of a spontaneous emitter inside or near a cavity. It can be calculated by dividing the power emitted by a dipole source in the presence of our resonant nanoparticles chain by the power emitted by the dipole in a homogeneous environment (without the resonant structure) since the emission rate is proportional to the local density of optical states (LDOS), and the LDOS is proportional to the power emitted by the source.

​​In the commercial software Ansys Lumerical FDTD, which we use for the modeling, A_eff can be expressed as dipolepower(f)/sourcepower(f), where the function dipolepower returns the power actually radiated in the environment (with the resonant nanoparticles chain), and the function sourcepower returns the power that would be radiated by a dipole in a homogeneous medium. This ratio is equal to the decay rate enhancement, which is the same as Purcell factor or A_eff in our case. More details about these functions and general simulation approach can be found in the Lumerical Knowledge Base (see https://optics.ansys.com/hc/en-us/articles/360041612594-Purcell-factor-of-a-microdisk).

Text added: (p. 4, line 130)  For Purcell effect enhancement definition we introduce a factor $A_{eff}=P/P_0$, where $P$ is radiated dipole power in the presence of the RWS and $P_0$ is the power that the same dipole would radiate in vacuum since the emission rate is proportional to the local density of optical states (LDOS), and the LDOS is proportional to the power emitted by the source.

Round 2

Reviewer 2 Report

The authors have carried out minor changes with respect to the rejected version. I understand and value their reasons for presenting almost the same manuscript. However, I must recommend to reject this manuscript.

I would be very pleased to accept a manuscript containing preliminary experiments, or the current manuscript in a specialized journal (computer science, numerical calculi, etc).

Kind regards